# A morphology-based machine learning model for scoring epithelial-mesenchymal plasticity using organelle dynamics

Justin Slager [1,5], Francesca Gatto[1,5], Benjamin Frey [2], Wenyang Shi[1], Bartlomiej Porebski[3,4], Jordi Carreras-Puigvert[2,4], Malgorzata Maria Parniewska[1] & Jonas Fuxe [1] ✉

Re-activation of epithelial–mesenchymal transition (EMT), a key developmental process, contributes to cancer progression and therapy resistance. Modulating EMT could be attractive as a therapeutic strategy, but there is a lack of methods that can quantify EMT states, including hybrid phenotypes. Here, we developed a morphology-based machine learning approach to score EMT based on changes in organelle dynamics. Using the Cell Painting assay and high-throughput microscopy, we trained a histogram gradient boosting classifier to identify stage-specific organelle remodeling during a time course of TGF-β1-induced EMT in mammary epithelial cells. The model achieved robust performance across datasets, capturing EMT kinetics, hybrid states, and reversal by mesenchymal–epithelial transition (MET). Importantly, the method accurately scored EMT in human breast cancer cells and lung cancer cells undergoing hypoxia-induced EMT, demonstrating cross-species, cross-inducer, and cross-cancer applicability. The results establish organelle morphology profiling as a scalable framework for quantifying epithelial-mesenchymal plasticity. The method offers a platform for drug discovery and identifying strategies to overcome EMT-associated resistance.

Cancer progression into metastatic disease is the leading cause of cancer-related mortality, accounting for over 90% of deaths in patients with solid tumors. Current therapies, including chemotherapy and immunotherapy, have limited efficacy in metastatic settings due to tumor heterogeneity and the ability of metastatic cells to evade immune responses. These challenges highlight the urgent need to develop novel therapeutic strategies specifically targeting the metastatic cascade.

A critical driver of cancer metastasis is cell plasticity, referring to the ability of tumor cells to dynamically transition between different phenotypic states. The epithelial-mesenchymal transition (EMT), a developmental cellular program, is a plastic process that is reactivated in cancer and that enables epithelial cells to acquire mesenchymal traits, including enhanced migratory and invasive capabilities. EMT and its reverse process, mesenchymal-to-epithelial transition (MET), are manifestations of epithelial-mesenchymal plasticity (EMP), which is activated in cancer progression and facilitating metastasis and tumor recurrence[1,2]. However, EMP is not a binary switch but instead, tumor cells may exhibit intermediate/hybrid EMT states, displaying a mixture of epithelial and mesenchymal traits, which complicates the classification of EMT[3,4]. Such hybrid states contribute to tumor heterogeneity, allowing cancer cells to adapt to various microenvironments during metastasis. Transforming growth factor-β (TGF-β) is the most potent and best characterized inducer of EMT and its expression is associated with poor prognosis and cancer progression[5,6]. Moreover, EMT has been implicated in the development of resistance to chemotherapy and immunotherapy, further complicating the treatment of metastatic disease and making EMT an attractive target for therapeutic interventions[6–8].

Despite the importance of EMP in cancer progression, existing methods to score EMT/MET transitions are limited. Traditional approaches rely on the expression of individual molecular markers, such as E-cadherin, N-cadherin, and Vimentin, and EMT transcription factors like members of the SNAIL, ZEB, and TWIST families[9]. While these individual proteins are useful EMT markers, their expression may be context-dependent, vary across cell types, and not sufficient in identifying different states of hybrid EMT[3,10]. RNA sequencing, although providing a more comprehensive view on

[1]Division of Pathology, Department of Laboratory Medicine, Karolinska Institutet, Stockholm, Sweden. [2]Department of Pharmaceutical Biosciences and Science for Life Laboratory, Uppsala University, Uppsala, Sweden. [3]Division of Genome Biology, Department of Medical Biochemistry and Biophysics, Science for Life Laboratory, Karolinska Institute, Stockholm, Sweden. [4]Department of Medical Biochemistry and Biophysics, Chemical Biology Consortium Sweden, Science for Life Laboratory, Karolinska Institutet, Stockholm, Sweden. [5]These authors contributed equally: Justin Slager, Francesca Gatto. ✉e-mail: jonas.fuxe@ki.se

the transcriptional reprogramming events that drive EMT, and functional assays, are time-consuming, expensive, and lack the scalability needed for high-throughput applications[9,11]. These limitations highlight the need for methods that are not dependent on the quantification of gene expression and that would rather take advantage of common features of EMT cells.

Emerging research shows that the EMT process involves complex subcellular changes at the level of organelles, including cytoskeletal rearrangements, mitochondrial fragmentation, and metabolic reprogramming, and changes to the endoplasmic reticulum, ribosomal biogenesis, and Golgi apparatus[12–17]. Together, these data suggest that organelle remodeling might serve as an indicator of EMT status. To explore this, we developed a method to analyze changes in organelle morphology by using the Cell Painting assay and high-throughput microscopy. Cell Painting is an established technique to image organelles through multiplexed staining by using fluorescent dyes that label intracellular structures, including the nucleus, mitochondria, endoplasmic reticulum, Golgi, and cytoskeleton[18–20]. Machine learning built on a histogram-gradient boosting classifier was used to train a model to predict EMT status at the population level, based on morphological profiles. We found that the method efficiently captured EMT dynamics across a spectrum of cells and EMT states, including hybrid phenotypes and MET. The results suggest that the method has potential for various applications, including drug screening for EMT modifiers, which could lead to the discovery of novel anti-cancer therapeutic strategies.

## Results

### Development of an EMT prediction model based on organelle dynamics

A phenotypic, high-throughput microscopy screening approach was established to model changes in organelle dynamics during TGF-β1-induced EMT in NMuMG mammary gland epithelial cells, a frequently used cellular model of inducible EMT[21,22] (Fig. 1). Cells were either left untreated (control) or exposed to TGF-β1 for 24 h, 48 h, 4 d, or 8 d. Morphological changes indicative of EMT, including cellular elongation and loss of cell-cell adhesion were gradually more evident during the time course of TGF-β1-induced EMT (Supplementary Fig. 1a). The expression of E-cadherin was downregulated while vimentin was induced (Supplementary Fig. 1b, c). Control cells and TGF-β1-stimulated NMuMG cells were stained with the Cell Painting kit to capture morphological profiles at each time point. The staining procedure resulted in prominent visualization of cellular organelles (Fig. 2 and Supplementary Fig. 2). Images were captured and imported into Cellprofiler for segmentation based on cellular, cytoplasmic, and nuclear content, and subjected to analysis through machine learning models to identify organelle changes related to EMT kinetics and states.

To model organelle dynamics during the time course of TGF-β1-induced EMT, three biological replicate 96-well plates were stained and imaged (Fig. 3a). This process yielded at least 100,000 single-cell profiles per condition, per plate, each capturing 2608 morphological output features, including measures of cell and organelle size and shape,

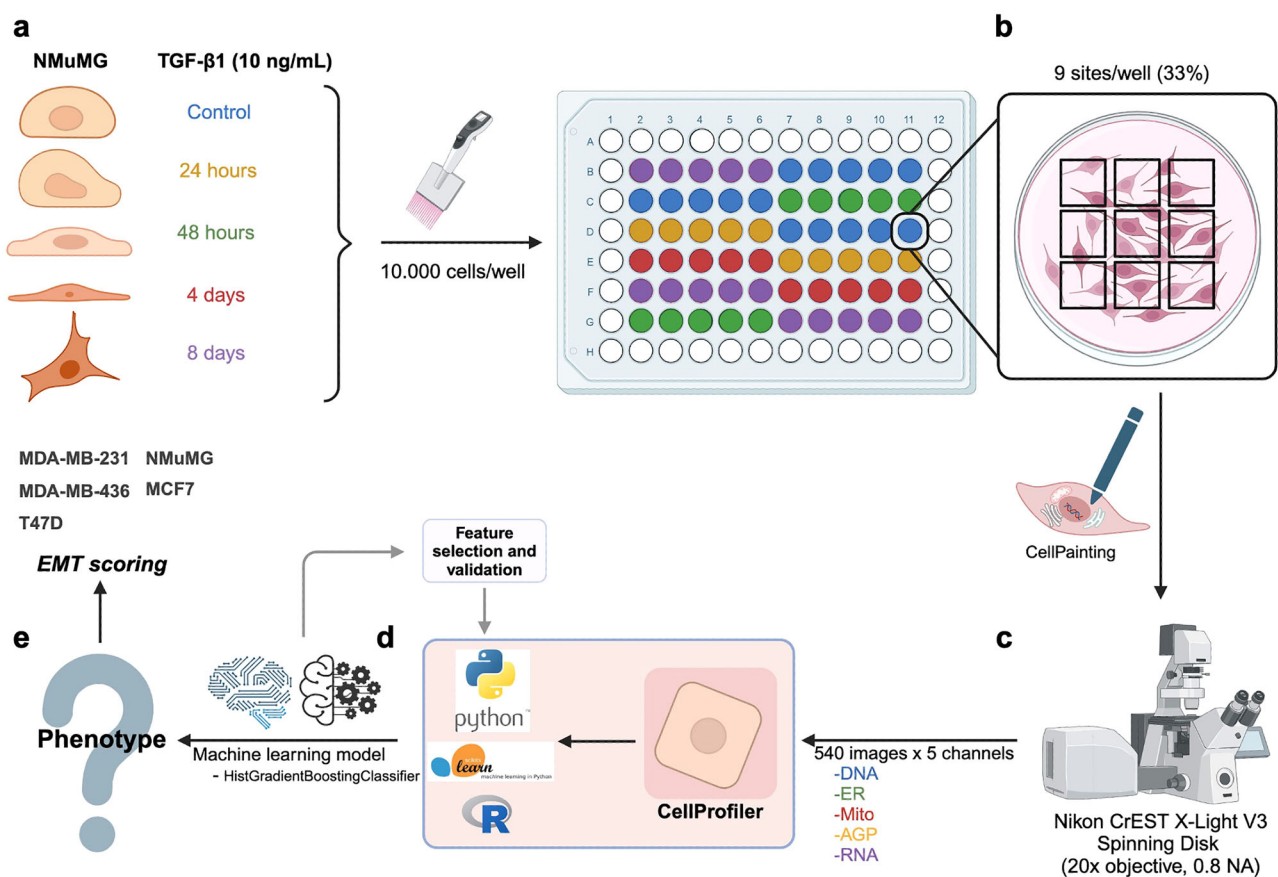

**Fig. 1 | Integrative workflow for EMT status prediction.** From bench to model, schematic overview of the approach taken to develop an EMT status prediction model and scoring method based on organelle dynamics and machine learning. **a** An established cellular model of TGF-β1-induced EMT in NMuMG cells was used to induce different stages of EMT by exposing cells to TGF-β1 for 24 h, 48 h, 4 d or 8 d. **b** Cells were seeded in 96-well plates and stained with the Cell Painting assay to visualize cellular organelles. **c** Images were captured through high throughput spinning disk microscopy and imported into CellProfiler for analysis and subcellular segmentation. **d** A machine learning model based on training of a histogram gradient boosting classifier was used to identify features predicting organelle morphology and phenotypic properties at each stage of TGF-β1-induced EMT. **e** An EMT scoring method was generated based on the top 15 identified morphological features and was shown to accurately, and annotation-independently, predict the EMT status of human breast cancer cells. Images were created with BioRender.com.

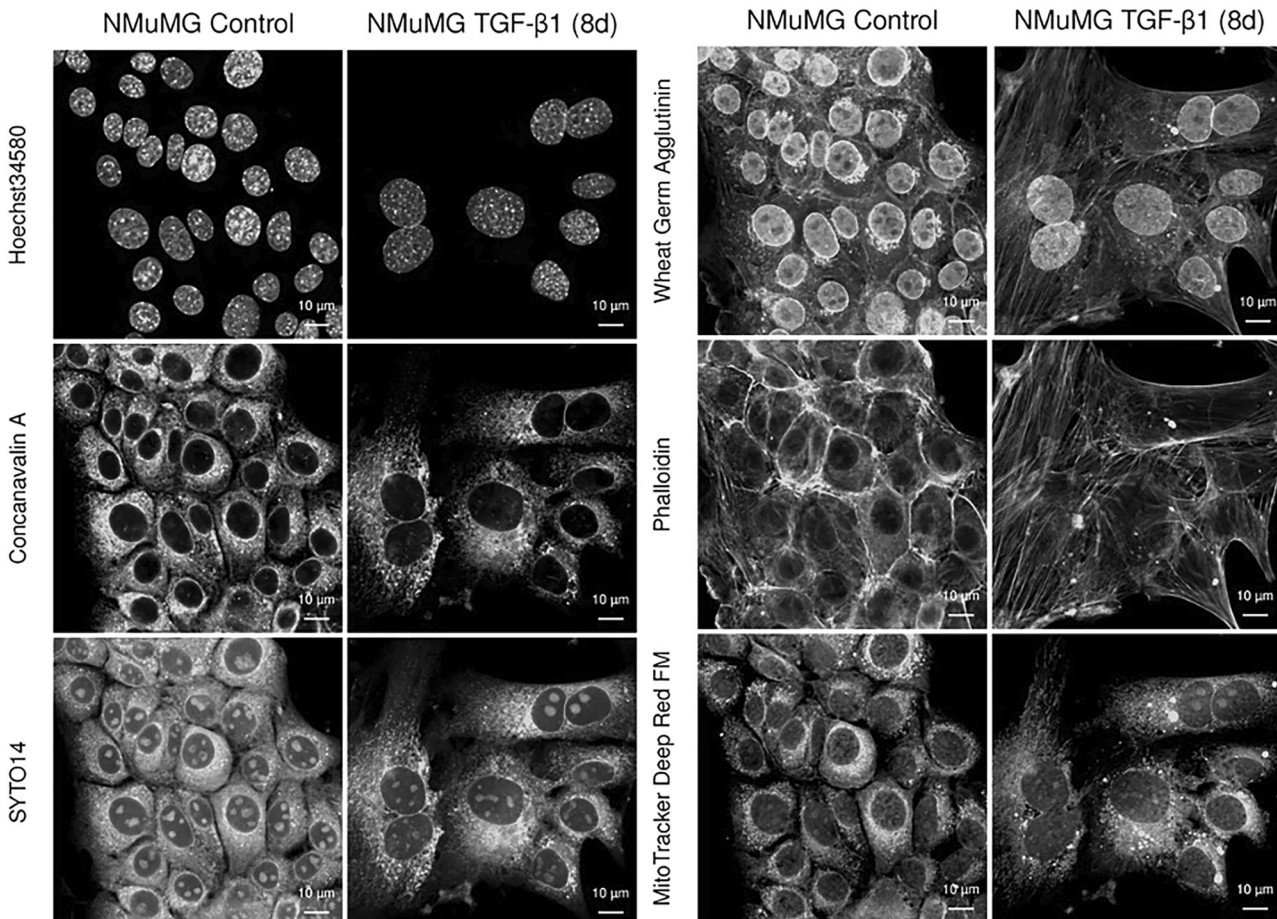

**Fig. 2 | Visualization of organelle changes during TGF-β1-induced EMT in NMuMG cells.** Fluorescent images of NMuMG cells stained with the Cell Painting kit according to protocol V3.5. to visualize cell nuclei (Hoechst34580), endoplasmic reticulum (concanavalin A), nucleoli and cytoplasmic RNA (SYTO14); golgi and plasma membrane (wheat germ agglutinin), F-actin cytoskeleton (phalloidin) and mitochondria (mitotracker deep Red). NMuMG cells were either left untreated (control) or treated with TGF-β1 for 8 d.

intensity and intensity distribution, texture, granularity, and colocalization/overlap across five fluorescent channels capturing the nucleus, mitochondria, endoplasmic reticulum, Golgi, and actin cytoskeleton. Single-cell profiles within each field of view were aggregated, resulting in 90 population-averaged (median) profiles per condition, per plate. This way, the model was developed based on population-averaged profiles rather than single cells to minimize biological variability within conditions, segmentation errors, and technical artifacts.

The datasets were normalized using the robustize method, which relies on the median and interquartile range (IQR) to minimize the impact of outliers and to balance feature contributions. Unlike Z-score normalization, which emphasizes high-variance features, or MAD robustize normalization, suited for single-cell data, the robust scaler is well-suited for aggregated profiles with features showing variability.

The first classifier was trained on 405 out of 2608 features (Fig. 3a and Supplementary Table 1), which were selected after data processing directed to the removal of 2203 features with low variance, high correlation, NA values, and other redundant characteristics to refine the data for modeling. UMAP visualization confirmed each phenotype as a distinct cluster, highlighting progressive changes and revealing the 4-day stimulation as a transition point, where cells shifted either towards the mesenchymal 8-day phenotype or were in closer proximity to the 48-hour timepoint (Fig. 3b).

The dataset was randomly divided into training and test sets to enable rigorous training and evaluation of multiple classifiers. A diverse range of classification algorithms were tested to ensure proper testing and generalizability (Fig. 3c). Models were trained and cross-validated on the training set using a Stratified Fold approach (5 folds, $N = 3$) with shuffling, resulting in average accuracy scores of 94–98% across classifiers. To identify the most predictive features, a permutation importance test ($N = 5$, with shuffling) was performed on the training set, measuring the change in accuracy after permuting each feature. Features with importance values above zero were further filtered by removing highly correlated features (>75% correlation) to eliminate redundancy, reducing the feature set from 25 to 15 (Supplementary Fig. 3). The refined feature set was used to retrain the classifiers, achieving average cross-validation accuracy scores of 93–96% and test set scores of 94–98% (Fig. 3c and Supplementary Table 2), indicating optimal model performance and generalizability.

The Histogram-Based Gradient Boosting Classifier model (Hist Gradient Boosting) was chosen for further testing and validation due to its scalability, efficiency with large datasets, and ability to handle non-linear relationships. Compared to the other models it offers improved computational efficiency, robust handling of high-dimensional data, including single-cell data, reduced sensitivity to outliers, and built-in regularization to minimize overfitting, making it well-suited for accurate classification of EMT states (Fig. 3c).

The final classifier was trained on the top 15 non-redundant features, which explained 45–50% of the variance between EMT states, after filtering out the redundant features (Fig. 4 and Supplementary Fig. 4). Among top predictive features were AreaShape_Solidity_cells, Texture_Correlation_CorrER_5_02_256_cytoplasm and Granularity_2_CorrMito_cells, which demonstrates the capacity of the method to identify changes in cellular shape, and in the structure of ER and

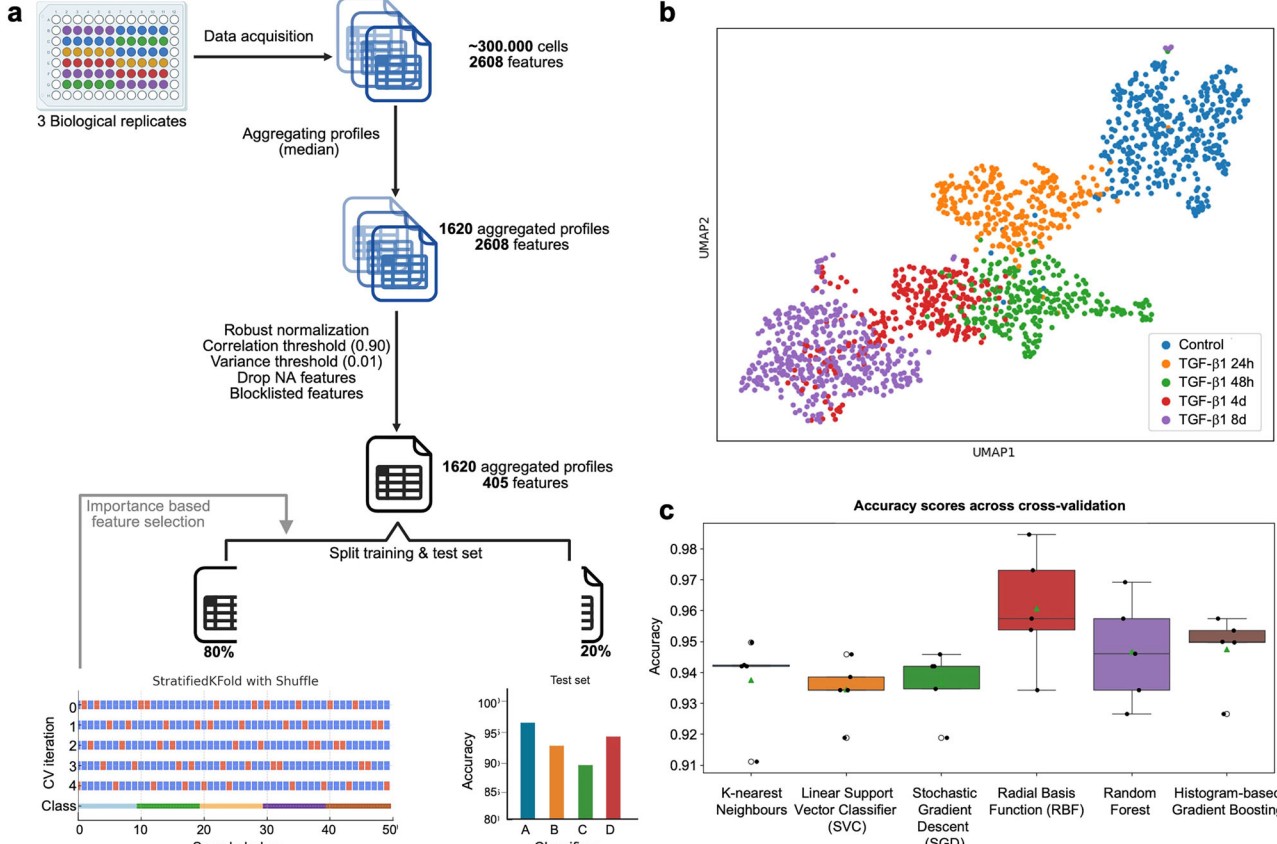

**Fig. 3 | Pre-processing of data and training of the EMT prediction model.**
**a** Overview of the data processing workflow. Three biological replicate 96-well plates, each containing five time points after TGF-β1 stimulation, were stained and imaged with the Cell Painting kit. This yielded 100,000 single-cell profiles per condition, per plate (300,000 cells in total), each capturing 2608 morphological output features. Single-cell profiles were aggregated in population-averaged (median) profiles per condition, per plate, resulting in 90 aggregated profiles (9 fields of view per well, times 10 wells per condition) per condition, per plate (1620 aggregated profiles in total). The datasets were normalized using the robustize method and features were removed based on low variance, high correlation, NA values, and other redundant

characteristics to refine the data for modeling. The resulting dataset was split into training and test sets (80/20 split), and a stratified 5-fold cross-validation with three repeats was performed. Images were created with Biorender.com. **b** UMAP visualization showing distribution of aggregated profiles in distinct clusters across time points during TGF-β1 stimulation. **c** Comparison of accuracy scores across different classifiers during cross-validation. Boxplots summarize accuracy scores across cross-validation folds, with individual fold accuracies represented as points. Box plot boundaries are 75th and 25th percentiles, with whiskers at ±1.5-fold of the inter-quartile range or the highest or lowest data point. The median is represented as a line in the boxplot, whereas the mean is denoted as a green triangle.

mitochondria, respectively, as features that could differentiate each time point based on the state of EMT. These were the criteria used to score and classify hybrid states during TGF-β1-induced EMT in NMuMG cells.

## Developing an EMT scoring method based on the prediction model

To validate our model, an independent experiment was performed with TGF-β1-induced EMT in NMuMG cells for the five time points. The data from this experiment were processed following the same pipeline as the model training dataset, and condition labels were withheld during the analysis. The model was used to predict the probability that cells in each well belonged to one of the five conditions (Fig. 5a and Supplementary Fig. 5).

To develop an EMT scoring method based on the prediction model we used previously established EMT scoring systems based on gene expression[12] as a reference tool for weighting the probability calculations. To do this, we re-analyzed RNA-sequencing data from a previous time course study of TGF-β1-induced EMT in NMuMG cells[21] by established gene expression-based EMT scoring systems. We used the established EMT hallmark gene set from Gene Set Enrichment Analysis (GSEA) to score EMT at the different time points of TGF-β1-induced EMT in NMuMG cells. The geometric mean expression values of 122 genes of the EMT Hallmark gene set were gradually induced during the time course of the experiment (Fig. 5b), and their mean fold induction was multiplied with the probability

scores for each time point during TGF-β1-induced EMT to develop an EMT scoring method (Fig. 5c).

In comparison, using the KS scoring method, we found that TGF-β1 stimulation for up to 36 h was associated with a more epithelial state, while stimulation for 48 h or longer resulted in a mesenchymal-dominant score (Supplementary Fig. 6a). Thus, this method could be used to measure EMT changes during the time course but was not capable of distinguishing the different time points from each other. Similar results were obtained using the 76GS scoring method, which is based on the expression of 76 genes normalized to *CDH1* (76GS), and which showed a gradual decrease of the epithelial phenotype over time (Supplementary Fig. 6b).

## Using the scoring method to capture EMT and MET

Using our EMT scoring method, we calculated EMT scores for each of the time points during TGF-β1-induced EMT in NMuMG cells. The results showed significantly increasing EMT scores at each time point during the time course (Fig. 5d). The scores for each time point were significantly different from all other time points, indicating that the method could differentiate between different hybrid stages of TGF-β1-induced EMT in NMuMG cells. To further evaluate the versatility of the model, we analyzed its capacity to score MET. NMuMG cells were treated with TGF-β1 for 8 days to induce EMT, after which TGF-β1 was withdrawn for increasing durations (2, 4, or 8 days) (Supplementary Fig. 7). The EMT scores were

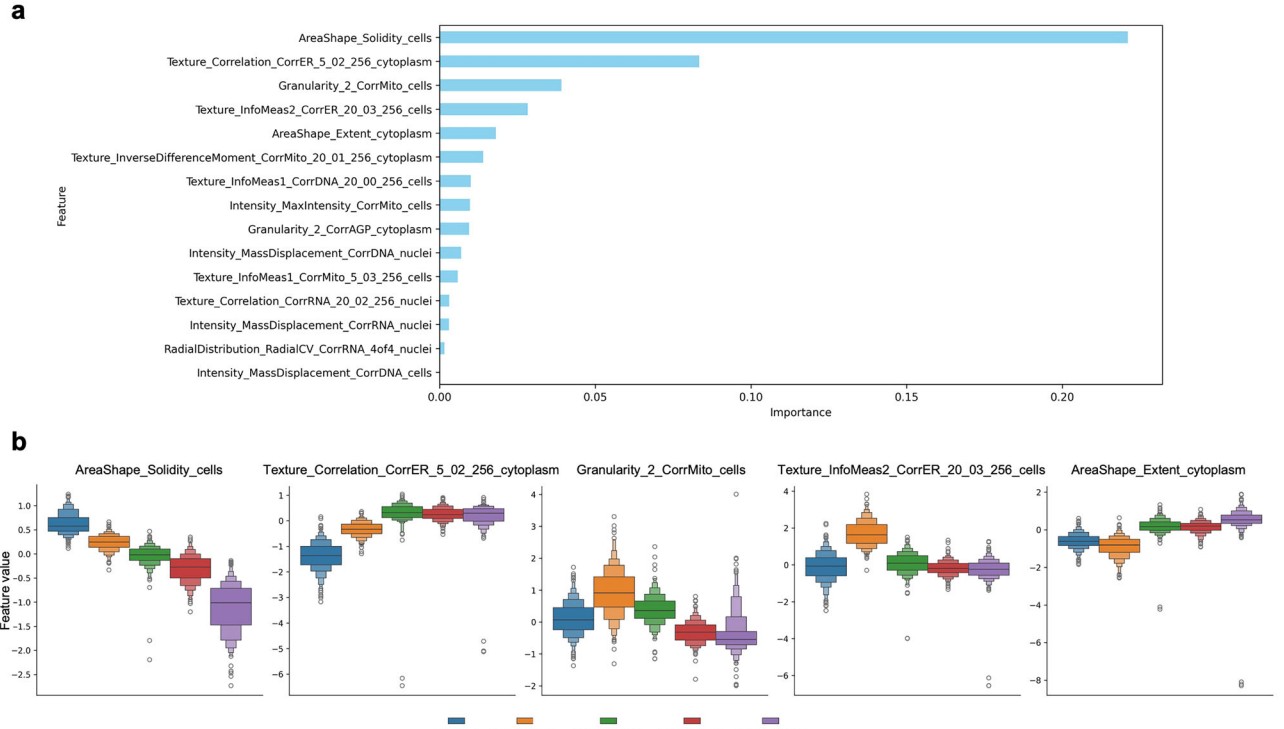

**Fig. 4 | Distribution of top features across different stages of TGF-β1-induced EMT. a** Feature importance was calculated by performing a permutation-based importance test, the average importance of each of the top 15 features is displayed. **b** Boxenplots depict the distribution of the top five features, determined by importance-based feature selection, across all experimental conditions. Each plot represents a feature, and each feature corresponds to a specific cellular or subcellular property, such as area shape, granularity, texture, intensity, and radial distribution. Each box plot visualizes data by showing the interquartile range, median, and progressively narrower percentiles, providing detailed insight into the distribution, especially in the tails, revealing subtle changes in feature values over time as cells undergo TGFβ1-induced EMT. These plots capture the progressive phenotypic alterations, with outliers represented as individual points beyond the plot range.

determined for each condition to assess the extent of EMT reversal. The results showed that after 2 days upon TGF-β1 withdrawal, the EMT score remained unchanged compared to 8 days after TGF-β1 exposure (Fig. 6a). However, at 4 days and 8 days after TGF-β1 withdrawal, EMT scores were gradually decreasing. These findings highlight the model's ability to capture the dynamic and reversible nature of EMT.

### Annotation-independent scoring of human breast cancer cells using the EMT prediction model

We then set out to explore whether the method could be used in annotation-independent scoring of EMT in human breast cancer cells. For this purpose, we selected breast cancer cell lines previously classified as either luminal/epithelial (MCF7 and T47D) or basal/mesenchymal (MDA-MB-231 and MDA-MB-436)[23]. Cell Painting staining and image analysis were performed for each of the cell lines followed by prediction modeling. The results showed significant differences in EMT scores between all the cell lines, with T47D cells having the lowest score followed by MCF7 cells, while MDA-MB-231 and MDA-MB-436 cells displayed higher EMT scores (Fig. 6b).

The scores were compared with established gene expression-based EMT scoring systems, including the EMT hallmark, KS, and 76GS scores, using gene expression data from 52 human breast cancer cell lines[23]. Gene expression-based analysis of the EMT hallmark revealed that T47D cells exhibited some of the lowest EMT hallmark scores among luminal breast cancer cells, whereas MCF7 cells displayed among the highest scores within this group (Fig. 6c). In contrast, MDA-MB-231 cells had higher scores compared to T47D and MCF7 but were among the lowest within the basal cell line category. Conversely, MDA-MB-436 cells demonstrated one of the highest EMT hallmark scores in the basal cell line group. Using the KS method, MDA-MB-436 and MDA-MB-231 cells showed high EMT scores,

while T47D and MCF7 cells exhibited low scores (Supplementary Fig. 7a). When the 76GS method was applied, similar trends were observed, with more pronounced differences among the four breast cancer cell lines compared to the KS method (Supplementary Fig. 7b). This highlighted that the relative ranking of EMT scores across human breast cancer cell lines was consistent between our organelle dynamics-based EMT scoring method and established RNAseq-based approaches.

### Applying the EMT scoring method in hypoxia-induced EMT in A549 lung cancer cells

To explore whether our model could also capture EMT induced by cues other than TGF-β1, we applied it to A549 lung cancer cells exposed to hypoxia, either alone or in combination with TGF-β1. Western blot analysis demonstrated decreased expression of E-cadherin and a tendency to increased levels of vimentin upon TGF-β1 stimulation, but not significantly after hypoxia (Supplementary Fig. 9). Cell Painting analysis revealed pronounced morphological remodeling under both conditions, with cells displaying elongated shapes compared with normoxic controls (Fig. 7a). Predicted probabilities and EMT scores showed that hypoxia and TGF-β1 individually induced EMT to a significant extent, with TGF-β1 being the strongest inducer (Fig. 7b, c). The combination of TGF-β1 and hypoxia resulted in a similar EMT score as TGF-β1 alone.

### Universal EMT scoring

Finally, we integrated EMT scores across all cell types and conditions investigated in this study to visualize the universal applicability of the model. Box-and-whisker plots positioned NMuMG cells at different TGF-β1 time points, breast cancer cell lines of luminal and basal origin, and A549 lung cancer cells under hypoxia or TGF-β1 treatment along a heterogeneous EMP axis (Fig. 8). This representation illustrates that the

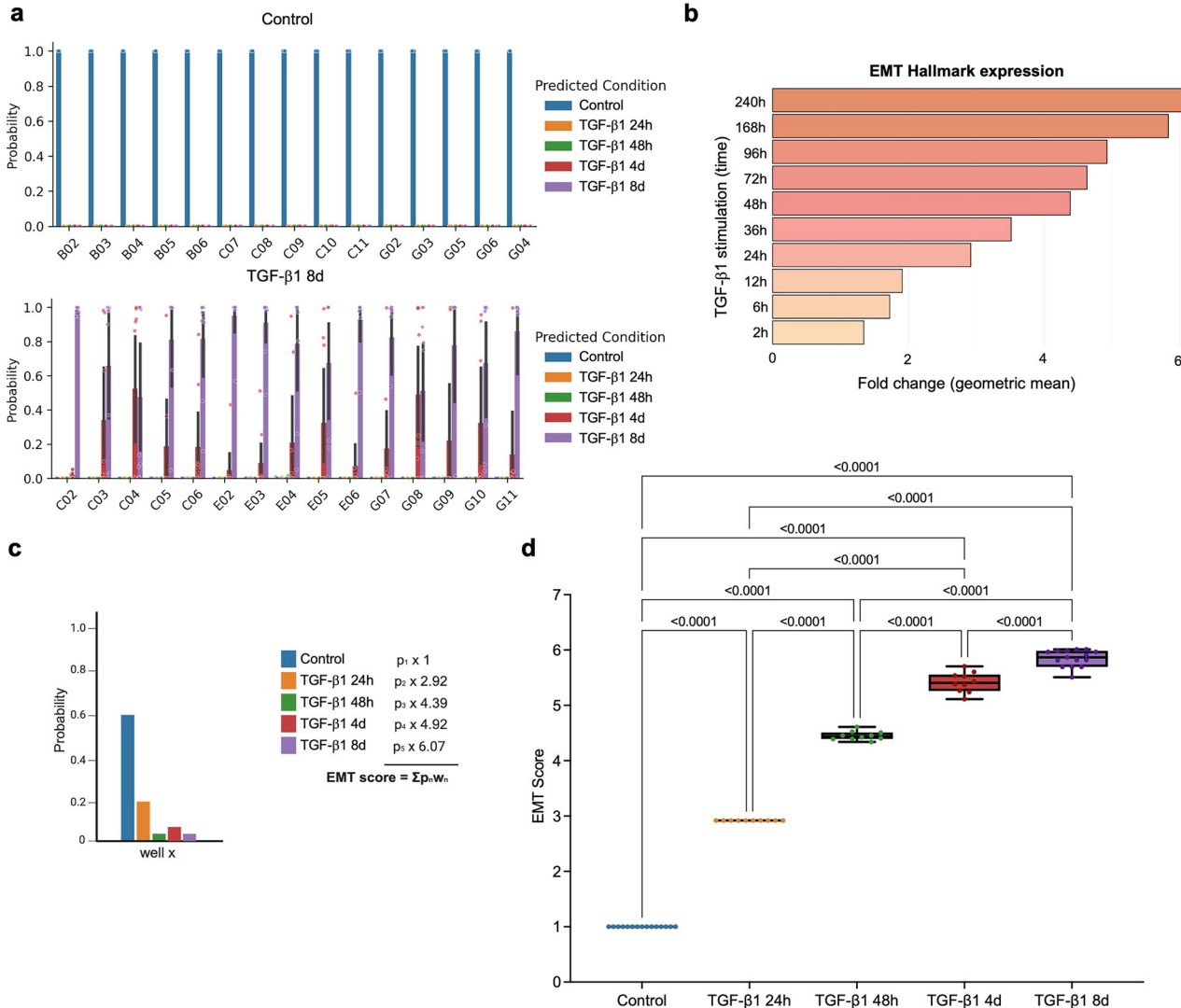

**Fig. 5 | Development of a scoring method for predicting EMT stages. a** Probability distribution for aggregated profiles in individual wells of NMuMG cells left untreated or stimulated with TGF-β1 for 8 d. The *x*-axis denotes well IDs, and the *y*-axis represents the probability (0–1) of cells in each well to belong to the predicted condition. Bars are color-coded as indicated. Error bars indicate variability (95% confidence interval) in the predictions (*N* = 9 per well). **b** The geometric mean expression of the Hallmark EMT gene set was plotted against the re-analyzed RNA-seq data from Meyer-Schaller et al.[21] from NMuMG cells treated with TGF-β1 for indicated time points. **c** Exemplified bar graph showing how the EMT scoring was developed. Bars represent the probability that well *x* belongs to each experimental condition (Control, TGF-β1 24 h, TGF-β1 48 h, TGF-β1 4 d, TGF-β1 8 d). To compute an EMT score, each probability ($p_n$) was multiplied by a condition-specific weight ($w_n$) and the products were summed (EMT score = $\Sigma\, p_n w_n$). Weights were defined as the fold change in EMT hallmark expression at each time point (shown in **b**), relative to the control, thereby linking classification probabilities to the measured degree of EMT induction. The equation used for scoring is listed in material and methods. The exemplified bar graph was created with BioRender.com. **d** Using the EMT scoring method, EMT scores were calculated for NMuMG cells at each time point during TGF-β1-induced EMT and were found to increase progressively during the time course, showing significant differences between all conditions. Box plot boundaries are 75th and 25th percentiles, with whiskers at ±1.5-fold of the inter-quartile range or the highest or lowest data point. The median is represented as a line in the boxplot.

EMT scoring method can discriminate between subtle states within the epithelial–mesenchymal spectrum and is capable of unifying diverse models of EMT into a common analytical framework. By providing a continuous scoring system that spans inducers, species, and cancer types, these results highlight the potential of organelle dynamics-based EMT scoring as a broadly generalizable tool for studying EMP.

## Discussion

In this study, we developed a morphology-based machine learning model that is capable of scoring EMT in cells by quantifying organelle dynamics. By taking advantage of high-throughput microscopy and the Cell Painting assay, we demonstrated that cell shape and subcellular remodeling can be used to track EMT and MET across a continuum of states. This approach

captures not only the epithelial and mesenchymal extremes but also the intermediate, hybrid states that are increasingly recognized as central to cancer cell plasticity and therapeutic resistance[1–4].

Our model was trained on NMuMG cells undergoing TGF-β1-induced EMT, with morphological profiles collected at defined time points of induction. Although data were obtained at the single-cell level, we used aggregated profiles per condition to minimize the influence of variability, segmentation errors, and technical artifacts, thereby increasing robustness while preserving the dominant trends of EMT progression in the cell population. In line with previous studies, we observed that TGF-β1-induced EMT was accompanied by pronounced morphological alterations, such as elongation and cytoskeletal reorganization, which are consistent with prior descriptions of EMT-associated cell shape changes[24–26].

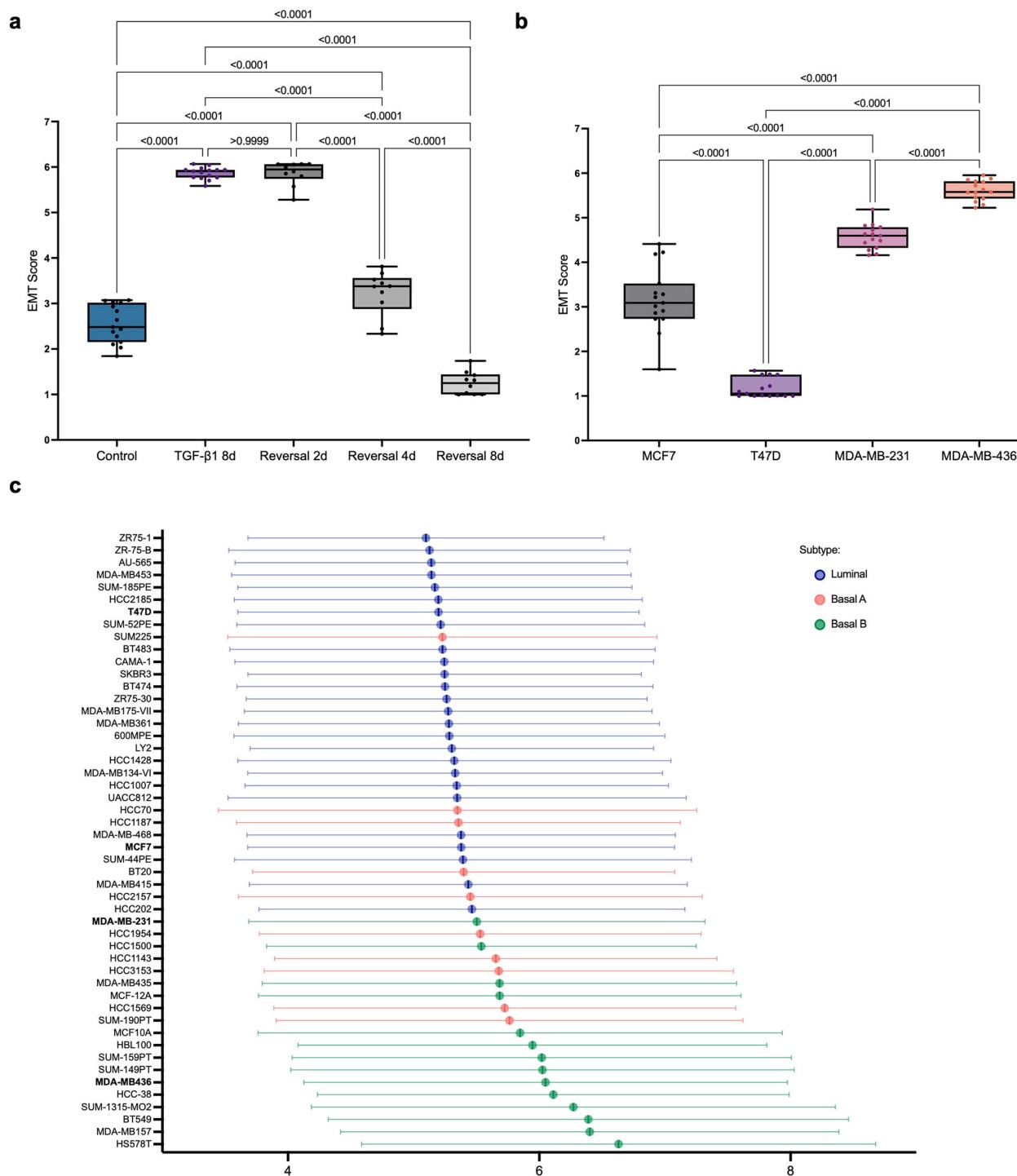

**Fig. 6 | Validation of the EMT scoring method.** The EMT scoring method was tested for its capacity to score reversal of EMT (**a**) and define the EMT status of human breast cancer cells in an annotation-independent fashion (**b**). **a** EMT scoring of NMuMG cells at different time points of reversal (2 d, 4 d, or 8 d) after TGF-β1-induced EMT for 8 d. Reversal was induced by withdrawal of TGF-β1 from the culture medium. At 2 d after TGF-β1 withdrawal, the EMT score remained high and at the same level as after TGF-β1-induced EMT for 8 d. At 4 d after TGF-β1 withdrawal, the EMT score was significantly decreased compared to TGF-β1-induced EMT for 8 d, but still higher compared to control. At 8 d after TGF-β1 withdrawal, the EMT score was decreased to a level, which was significantly lower than both TGF-β1-induced EMT for 8 d, the 4 d withdrawal time point and even lower than the unstimulated cells (control). **b** Annotation-

independent scoring of human breast cancer cells (MCF7, T47D, MDA-MB-231, and MDA-MB-436) using the EMT model. The results showed the lowest score for T47D cells, followed by MCF7 cells. Both of these are known as luminal breast cancer cells with epithelial characteristics. MDA-MB-231 and MDA-MB-436 cells, which are known as basal breast cancer cells with mesenchymal characteristics, were scored significantly higher. Box plot boundaries are 75th and 25th percentiles, with whiskers at ±1.5-fold of the interquartile range or the highest or lowest data point. The median is represented as a line in the boxplot. **c** Mean expression of EMT Hallmark genes in 52 human breast cancer cell lines based on the GOBO database. Cell lines are ranked from highest to lowest values and the cell lines used for scoring with the EMT model (MCF7, T47D, MDA-MB-231, and MDA-MB-436) are highlighted in bold.

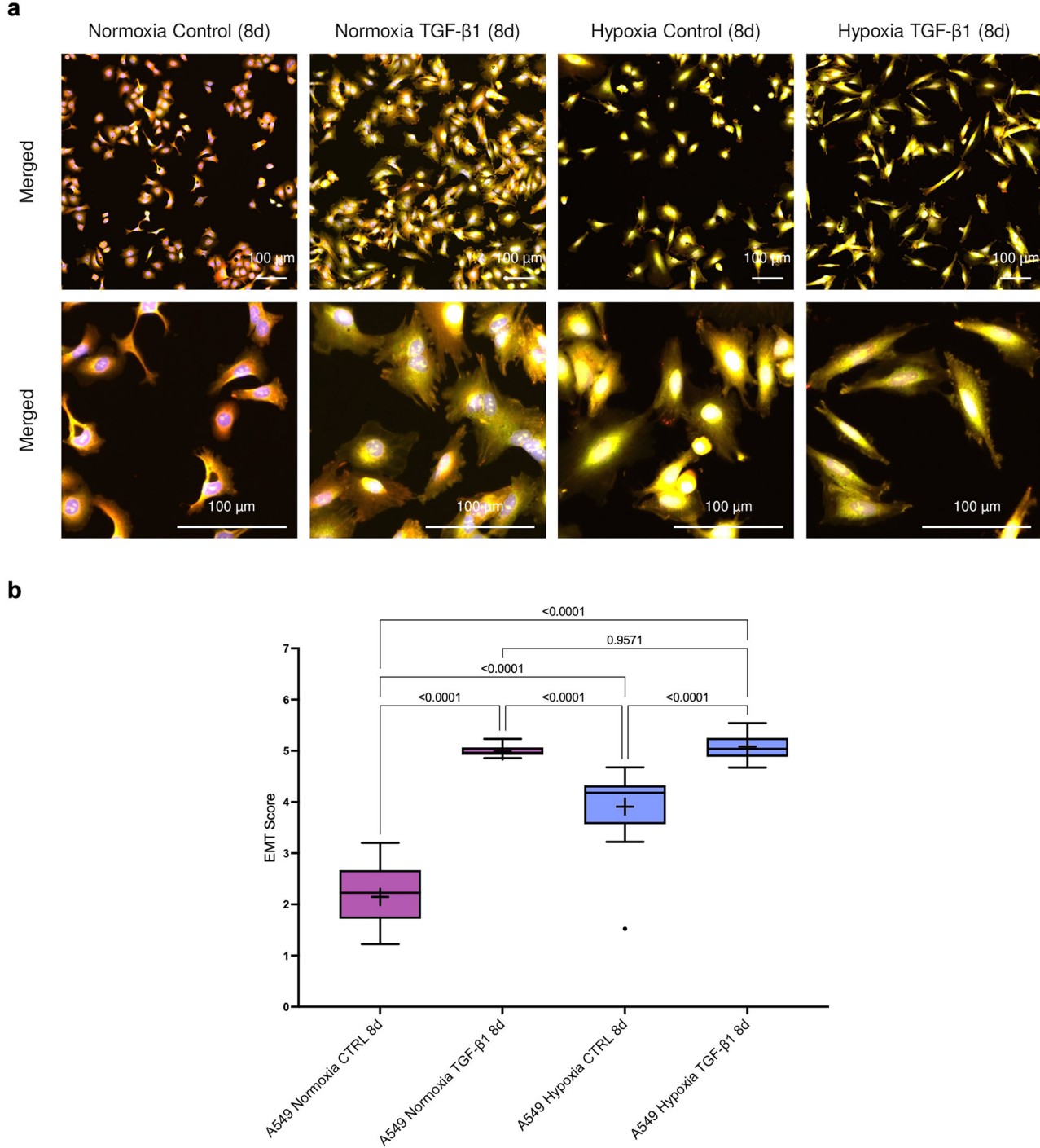

**Fig. 7 | Scoring of hypoxia-induced EMT in A549 cells. a** Fluorescent images of A549 cells (merged channels) cultured under normoxia or hypoxia with or without TGF-β1 treatment for 8 days and stained with the Cell Painting kit. Scale bars = 100 μm. **b** EMT scores for each condition, shown as Tukey boxplots (median, IQR, whiskers, outliers). Both hypoxia and TGF-β1 alone significantly increase EMT compared with controls. Exact p-values from statistical comparisons are shown.

From the full feature set, we identified a compact group of 15 features that were sufficient to discriminate between different EMT states. Among these, cell solidity emerged as the most influential predictor, reflecting changes in the degree of cellular shape irregularity and shape remodeling. Equally important were textural features derived from the endoplasmic reticulum and mitochondria, which pointed to organelle clustering and rearrangement, while nuclear chromatin and RNA distribution features captured transcriptional and architectural reorganization during the EMT process. These features have also been tightly linked to increased cell motility[27,28], supporting the notion that the organelle- and morphology-

based EMT states captured by our model may reflect functional motile phenotypes. Together, these findings indicate that EMT is accompanied by coordinated remodeling at subcellular levels, extending the biological understanding of plasticity beyond traditional marker-based definitions, and consistent with the known organelle remodeling that accompanies EMT[12–17]. These data point to important roles of various cellular compartments in cell plasticity, but whose biological relevance remains to be fully elucidated, and support the view outlined in the 2019 EMT International Association consensus[10], that EMT is best described as a continuum rather than a binary process.

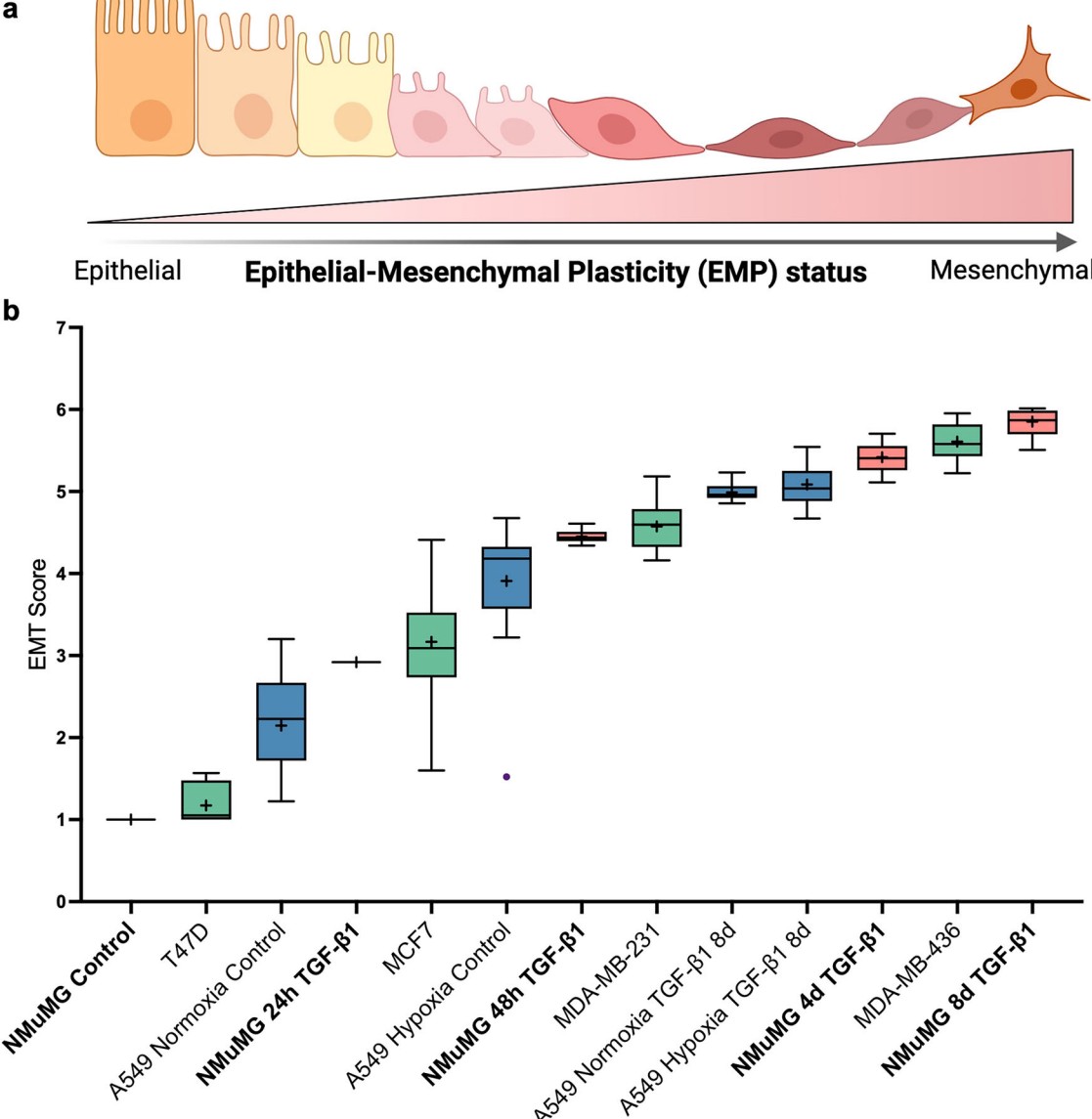

**Fig. 8 | Universal EMT scoring across cell lines and conditions. a** Schematic drawing showing EMP status related to EMT scores in cell lines used in the study. The illustration was created with BioRender.com. **b** Box-and-whisker plots (median, IQR, outliers plotted as individual points and mean values indicated by "+") showing the distribution of EMT scores across different cell lines and exposures to EMT-inducing conditions. Groups of cell lines are color-coded (red, NMuMG; green, breast cancer; blue, lung cancer) and positioned across an EMP axis.

The model identified EMT hybrid states, not by a few markers but by evolving, composite morphological signatures based on the top 15 features. This approach favored population-level profiles over detailed single-cell trajectories, thus prioritizing robustness, while limiting resolution of rare phenotypes. Previous work shows that EMT trajectories at the single-cell level can be diverse and variable[29,30], and studies have linked morphological heterogeneity to distinct motile properties[27,28]. In line with this, our results also highlighted the presence of cellular heterogeneity as although the overall EMT scores increased progressively with TGF-β1 stimulation, classifier probability distributions revealed that within the same condition, subsets of cells were predicted to belong to neighboring EMT states. This suggests that EMT does not occur synchronously across all cells but rather unfolds as a continuum of epithelial, intermediate, and mesenchymal-like phenotypes. Another explanation to the observed heterogeneity is that not all cells are equally responsive to EMT induction. In line with previous studies, we found a delay in the reversal of EMT, which was first observed at 4 days after TGF-β1 withdrawal[31]. At the 8-day reversal, the EMT score was lower than that of untreated controls, suggesting that cells reverting after prolonged EMT may not return to their exact original epithelial state. This might be explained by selection of particularly plastic subpopulations with growth advantage, or by re-entry into alternative phenotypic trajectories during the MET process. Future approaches may benefit from integrating aggregated and single-cell–based analyses to achieve both reproducibility and fine-grained insight.

Our method showed generalizability across contexts. Beyond TGF-β1-induced EMT, we found that the method could capture MET reversal, hypoxia-driven EMT in A549 lung cancer cells, and annotation-independent classification of luminal versus basal breast cancer cell lines. This indicates that subcellular remodeling is a conserved hallmark of EMP, and at least to some degree independent of the upstream inducing stimulus or cellular origin. Such generalizability is critical for studying tumor plasticity, as cancers are exposed to diverse EMT drivers, including growth factors, hypoxia, and inflammatory cues.

When compared to other existing EMT screening strategies, our approach offers some advantages. Marker-based systems are context-dependent and often insufficient to resolve hybrid states[9,10].

Transcriptomic-based scoring methods provide comprehensive information but are costly, time-consuming, and not easily scalable for high-throughput applications[11,32]. Previous morphology-based screens have primarily focused on a few descriptors, such as cell shape alone[24,28]. In contrast, our model uses an unbiased selection of features encompassing cell architecture and multiple organelle compartments, which improves sensitivity and resolution. Our method builds on widely accessible Cell Painting workflows[18–20], and is compatible with large-scale screening pipelines. This positions the method as a practical platform for systematic identification of EMT-modulating compounds, an unmet need in efforts to target metastasis and therapy resistance[33–37]. Although primary tumor cells can be more difficult to culture on 2D substrates due to limited adhesion or growth, these challenges are routinely addressed by adapting coatings (e.g., laminin, collagen) or growth conditions. Once established, patient-derived cells can be processed with the same staining and imaging workflow described here, enabling the method's use in preclinical EMT phenotyping and drug screening.

While promising, our study has limitations. First, the reliance on population-level profiles, while robust, may overlook rare transitional phenotypes that could be biologically relevant. Incorporating single-cell trajectory modeling[26,29,30] may provide complementary insights into fate bifurcations and plasticity dynamics. Second, although we identified predictive morphological features, their causal biological significance remains to be fully elucidated. For example, how mitochondrial granularity and ER clustering contribute to EMT remains an open question. Mechanistic studies dissecting organelle function during EMT may uncover novel therapeutic opportunities. Third, while a single, high dose of TGF-β1 over time spans a broad EMT continuum, steady-state cancer cell lines may occupy a narrower dynamic range. However, while this may limit absolute spread, our scoring method still improved discrimination compared to commonly used transcriptomic metrics, as exemplified by studying the established MCF7 and MDA-MB-231 breast cancer cell lines. Furthermore, our analysis showed that NMuMG cells, which are non-transformed, scored more epithelial than MCF7; this is consistent with the concept that many cancer lines reside in intermediate/hybrid EMT phenotypes. The observation that the scores increased from NMuMG < MCF7 < MDA-MB-231 along the EMT axis indicates that our calibration preserves biological ordering and captures epithelial headroom that cancer lines often lack.

Altogether, the findings establish organelle dynamics as a powerful and generalizable predictor of EMP states. By integrating high-content imaging with interpretable machine learning, our framework enables detection of EMT and MET transitions, hybrid states, and cellular heterogeneity across diverse biological contexts. Moreover, the platform provides practical utility for high-throughput screening and therapeutic discovery.

## Methods
### Cell culture
T47D and A549 cells were cultured in RPMI 1640 (#52400-025, ThermoFisher, Gothenburg, Sweden) supplemented with 1× MycoZap Plus-CL (#VZA-2012, Lonza Bioscience, Basel, Switzerland) and 10% heat-inactivated Fetal Bovine Serum (FBS) (#SV30160.03HI, Fisher Scientific, Gothenburg, Sweden). Namru Murine Mammary gland (NMuMG), MDA-MB-231, MDA-MB-436, and MCF7 were grown in DMEM (#41966-029, ThermoFisher Scientific) supplemented with 10% FBS and 1× MycoZap Plus-CL. All cells were from American Type Culture Collection (ATCC, LGC Standards—Nordic—UK Office, Middlesex, United Kingdom) and cultured in 25 or 75 cm$^2$ tissue culture flasks (#83.3910.002, Sarstedt, Nümbrecht, Germany) in a humidified atmosphere of 37 °C with 5% CO$_2$. The cells were passaged 2–3 times a week by adding TrypLE$^{TM}$ Express (#12605-028, ThermoFisher) for 5–10 min at 37 °C. All cell lines regularly tested negative for common mycoplasma contaminations before and during experiments (#rep-mysnc-50, InvivoGen, Toulouse, France). Moreover, a PCR-based Mycoplasma check was performed on all parental cell lines prior to experiments (Mycoplasmacheck, Eurofins Genomics, Ebersberg, Germany).

### Western blot
Cells were lysed in RIPA buffer (#89900, Thermo Scientific) supplemented with protease and phosphatase inhibitors (1:100, #78442, Thermo Scientific). Lysates were incubated on ice for 15 min and then centrifuged at $14{,}000 \times g$ for 15 min at 4 °C to remove debris. Protein concentrations were determined using a BCA assay (#23223 and #23224, Thermo Fisher Scientific). Equal amounts of protein (15 µg per lane) were mixed with 4XLDS sample buffer (#B0007, Invitrogen) + 5% 2-mercaptoethanol (#35602BID, Thermo Scientific), boiled for 10 min at 95 °C, and separated by SDS-PAGE on 4–12% polyacrylamide gels (#NW04122BOX, Invitrogen). Proteins were transferred onto nitrocellulose membranes (#1704159, BioRad) using a semi-dry transfer system (Trans-Blot Turbo, BioRad). Membranes were blocked in 5% nonfat dry milk in TBS-T (25 mM Tris-HCl pH 7.6, 150 mM NaCl, 0.1% Tween-20) for 1 h at room temperature, and then incubated overnight at 4 °C with primary antibodies diluted in blocking buffer. The following primary antibodies were used: E-cadherin (1:1000, #24E10, Cell Signaling Technology, Leiden, The Netherlands), Vimentin (1:1000, #5741, Cell Signaling Technology), and GAPDH (1:10.000 #MA5-15738-HRP, Thermo Fisher). After washing with TBS-T, membranes were incubated with HRP-conjugated secondary antibodies (1:10.000 goat anti-rabbit IgG #7074 and 1:10.000 goat anti-mouse IgG #7076, Cell Signaling Technology) for 1 h at room temperature. Immunoreactive bands were visualized using enhanced chemiluminescence (ECL) detection reagent (#WBLUF0100, Millipore) and imaged using a LI-COR Odyssey imaging system.

### EMT induction
In NMuMG cells, EMT was induced by 10 ng/mL of recombinant human Transforming Growth Factor-β1 (TGF-β1) (#7754-BH, R&D Systems, Abingdon, United Kingdom) for multiple time points (1, 2, 4, or 8 days). For the 4-day time point the cells were split once and for the 8-day time point twice to ensure a sub-confluent ( ~ 80%) state during the entire experiment. In A549 cells, EMT was induced either by stimulation with TGF-β1 (10 ng/mL), hypoxia, or a combination of both stimuli. For hypoxic conditions, cells were cultured in a humidified incubator (hypoxic chamber) at 37 °C with 5% CO$_2$ and 1% O$_2$. Both treatments were carried out for 8 days, during which cells were split twice to prevent overconfluence.

### Cell painting assay
Twenty-four hours prior to the cell painting assay, all cells were detached and seeded into 96-well plates (ibiTreat #1.5 polymer surface, #89606, Ibidi, Munich, Germany) at a density of 10,000 cells per well (see Supplementary Table 3 for plate layouts). The Image-iT$^{TM}$ Cell Painting Kit (#I65000, Invitrogen) assay was executed as described[18]. Staining concentrations were adopted according to Cell Painting version 3.5, an optimized protocol for robust outputs[38]. In brief, cells were live stained with 500 nM mitochondrial marker (MitoTracker$^{TM}$ Deep Red FM, #M22426, Thermofisher) staining solution for 30 min at 37 °C, 5% CO$_2$, and 95% relative humidity. Cells were fixed using 4% PFA (methanol free) and incubated for 20 min at RT. PFA was removed and after washing with HBSS (1X) replaced with the Cell Painting staining solution diluted in 0.1% Triton$^{TM}$ X-100 HBSS (#14065-056, Gibco$^{TM}$ Thermofisher) + 1% BSA according to Supplementary Table 4. The plates were incubated for 30 min at RT and washed 3 times for 5 min with 1× HBSS. After the final washing step, the wells were filled with 1× HBSS and sealed with an adhesive seal. The sealed plates were stored at 4 °C until imaging.

### EMT reversal
NMuMG cells were employed to evaluate the reversibility of EMT under various conditions. Cells were divided into different groups: a positive control group stimulated with 10 ng/mL TGF-β1 for 8 days, and an unstimulated control group. For the reversal studies, cells, after 8 days of TGF-β1 stimulation, were washed twice with DPBS and centrifuged (5 min, 500 RCF) to eliminate any residual TGF-β1, then cultured for additional periods of 2, 4, or 8 days in TGF-β1-free medium to assess the reversion to epithelial characteristics.

## Image acquisition

The image acquisition was performed on a Nikon CrEST X-Light V3 spinning disk microscope system with the 20X objective (NA = 0.8, WD = 0.8 mm, 0.33 μm/pixel). Lumencor despeckler lasers of various wavelengths (Supplementary Table 5) were used for exciting the different dyes. A Celesta X-light Pad multiband filter wheel was used for filtering the emission light. The emitted light was captured by a photometrics kinetix air-cooled sCMOS back-illuminated monochrome camera (95% QE) with 12-bit depth. The acquisition was automated by a pipeline in the JOBS module of the Nikon NIS elements software (version AR 5.30.02). In short, the control and most extreme conditions were manually checked and the exposure time per dye was set accordingly to prevent over- or under-exposure of the signals. Moreover, the Nikon Perfect Focus System (PFS) was enabled to automatically find the in-focus plane of view when switching from well to well. Per image, an extra auto-focus based on the endoplasmic reticulum (AF488) signal was enabled to ensure optimal in-focus for all 9 image sites per well. Images were captured starting from highest (Mito-Tracker) to the lowest (Hoechst) emission spectra and saved to an ND2 file combining all 9 images with 5 channels each, per well.

## Image processing

The acquired ND2 files were split into single-channel greyscale 12-bit depth TIF images using the NIS elements software. These images were analyzed using CellProfiler (version 4.2.5) software[39]. In the curated CellProfiler pipeline (see "Data availability"), the illumination correction modules were used to correct for the uneven illumination per image. The Hoechst staining was used to identify the nucleus, which functioned as the primary object. The secondary object was identified with the AGP channel and showed the outline of the cell. Afterwards the following measurements were performed on a single-cell bases: intensity, intensity distribution, colocalization, over-lap, texture, granularity, and size and shape of the different organelles. Data was saved both in a CSV file and in a MyQLite database for downstream processing. Three random images per condition were manually checked for accuracy of cell segmentation and the CellProfiler pipeline was run when accuracy was at least 90%.

## Cell painting data pre-processing and modeling

Data processing, statistical analysis, and machine learning modeling were performed using Python (version 3.12.4). Key libraries included Pandas (version 2.2.2) and NumPy (version 1.26.4) for data manipulation, SciPy (version 1.13.1) for statistical tests, and Scikit-learn (version 1.4.2) for machine learning. Visualizations were created using Matplotlib (version 3.8.4) and Seaborn (version 0.13.2). Data from three biological replicate plates were processed by loading cellular features from CSV files, including nuclei, cytoplasm, and whole-cell measurements. Each dataset was con-catenated horizontally after appending suffixes to distinguish feature origins. Missing values were removed during this step, ensuring clean and integrated datasets for downstream analysis. Cellular features from each plate were aggregated at the image level using the median function. Aggregation was performed across nuclei, cytoplasm, and whole cell features to produce representative profiles for each image per well. This step ensured robust feature representation for downstream analysis. Aggregated data from the three plates were preprocessed to remove missing values and normalized per plate using robust scaling based on the median and IQR per feature across conditions. Feature selection was applied to remove highly correlated (>0.9), low-variance (<0.01), and metadata-rich columns using Pycytominer (version 1.1.0). A blocklist was applied to remove specific features from the normalized and aggregated dataset. Features were excluded based on previous experience and predefined patterns associated with irrelevant or redundant measurements. The resulting dataset was con-catenated into a unified format for downstream analysis. UMAP embedding was computed using the Scanpy framework (version 1.10.3) to visualize the aggregated profiles. The embedding was initialized with a trajectory-informed layout derived from Partition-based Graph Abstraction (PAGA)[40], capturing transitions between experimental conditions. The

reason for using PAGA rather than UMAP alone, was that PAGA looks as how strongly groups of cells (aggregated profiles) are connected based on their feature similarity, preserving the global connectivity of the data across time. This approach resulted in a more integrated readout of EMT state.

To evaluate classifier performance, six machine learning models were tested: K-nearest neighbors, linear support vector classifier (SVC), Sto-chastic gradient descent, radial basis function kernel SVC, Random Forest, and Histogram-based Gradient Boosting. The dataset was split into training and test sets (80/20 split), and a stratified 5-fold cross-validation with three repeats was performed. For each model, cross-validation and test set accuracy scores were calculated. Classification reports, detailing perfor-mance metrics for both cross-validation and test phases, were saved to a CSV file (Supplementary Table 2). Feature importance was assessed using permutation-based importance on the training set. The Histogram-based Gradient Boosting classifier was used to evaluate the impact of individual features on model performance. Permutation-based importance scores were used to rank features, and the top 25 features were selected for further analysis. A Spearman correlation matrix was calculated for these features and strong correlations (absolute value > ±0.75) were excluded due to redundancy. The results provided insights into which features contributed most to the model's predictive performance. The trained histogram gradient boosting classifier and metadata (class names and feature names) were serialized using joblib (version 1.4.2) and saved as a .cpamodel file for reproducibility and future use (see "Data availability").

## EMT scoring

Datasets for the classification pipeline were preprocessed identically to those used for training and evaluating the model, ensuring consistency in feature scaling, transformation, and handling of missing values. To visually inspect for outliers, principal component analysis (PCA) was performed on the aggregated profiles, exclusively considering the top 15 features used in the classification model, grouped by condition (also for the control). Outliers were identified using an Isolation Forest algorithm with a contamination rate of 5%. Outlier detection was conducted separately for each condition, and profiles identified as outliers were removed. The resulting dataset was used for subsequent analyses. Prediction probabilities were calculated using the trained histogram gradient boosting classifier based on the 15 most important features. EMT scores were calculated by weighting condition probabilities with predefined progression scores with the following equa-tion:

$$\text{EMT Score} = P(\text{Control}) \cdot 1 + P(24\text{h TGF} - \beta1) \cdot 2.92 + P(48\text{h TGF} - \beta1) \cdot 4.39$$
$$+ P(4\text{d TGF} - \beta1) \cdot 4.92 + P(8\text{d TGF} - \beta1) \cdot 6.07$$

The predefined progression scores, which were used to weight the probabilities, were calculated as the fold change in the EMT hallmark score at each time point compared to the control (Fig. 5b).

## RNA-seq analysis

RNA sequencing files from deposit GSE112797 in the GEO database named "Kinetic analysis of TGFbeta-induced EMT in NMuMG/E9 cells" were re-analyzed. Analysis was performed according to the original paper with minor adjustments[21]. The raw count data was analyzed using the R package DESeq2 (version 1.42.0). Rows with a low mean gene count (>10) were filtered out. The DESeq2 model displayed significant differ-ential expressed genes compared to the factor level reference "0 h treat-ment". FDR-adjusted p-values and log2 fold changes (Log2FC) were used to plot the average Log2FC over treatment time of the genes present in the Hallmark EMT (Pubmed ID: 26771021), part of the Molecular Sig-nature Database (MSigDb). EMT scoring methods are derived from Chakraborty et al. 2020[11] and calculate the probability that an NMuMG hybrid stage is either epithelial or mesenchymal (KSScore) and the weighted sum of 76 EMT-related genes correlated to *CDH1* (E-Cadherin) to predict the correlation with the epithelial state (76GS score). The normalized expression of 51 human breast cancer cell lines in the GOBO

database from deposit E-TABM-157[41] was analyzed to calculate EMT scores using the average expression of the genes in the EMT hallmark list, the KSS and the 76GS scores, respectively.

## Statistics

Statistical analysis for comparisons between more than two groups was performed with a one-way ANOVA followed by Tukey's multiple testing correction using both Python (version 3.12.4) and the statistical software in GraphPad Prism (version 10.4.0).

## Reporting summary

Further information on research design is available in the Nature Portfolio Reporting Summary linked to this article.

## Data availability

Image data, numerical data, and uncropped western blot data used for analyses in the manuscript are available at the Swedish National Data Service (https://doi.org/10.48723/m1cg-v223). All other data are available from the corresponding author on reasonable request.

## Code availability

The codes, pipelines, and datasets used are available in the following GitHub repository: https://github.com/jonfux/Cell-Painting-EMT.

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

## Acknowledgements
This work was supported by the Swedish Cancer Society (#211739 Pj, #243842 Pj to J.F.), Radiumhemmets Research Funds (#211092, #231172 to J.F.), Karolinska Institutet Research Funds (#2024-03053 to J.F.) and the Robert Lundberg Memorial Foundation (project grant to F.G.). The funders had no role in study design, data collection and analysis, decision to publish or preparation of the manuscript. Furthermore, the authors would like to thank the Biomedicum Imaging Core (BIC) facility at Karolinska Institutet for providing access to imaging equipment and for their technical assistance during image acquisition and analysis.

## Author contributions
M.M.P. and J.F. conceptualized the study. M.M.P., W.S., B.P., F.G., J.S., and J.F. developed experimental methodology. J.S. and W.S. optimized imaging settings. J.S. and F.G. performed data acquisition, including imaging and data processing. J.S., F.G., and B.F. conducted formal analysis. J.S. developed the main code. B.F. subsequently validated and optimized the code. J.S., F.G., B.F., and J.F. prepared figures and data visualizations. J.S., F.G., and J.F. wrote the main manuscript text. J.C.P. and J.F. provided guidance throughout the study. All authors reviewed the manuscript.

## Funding

## Competing interests
The authors declare no competing interests.
