## [Transparent Peer Review file · Communications Biology]

A Morphology-Based Machine Learning Model for Scoring Epithelial-Mesenchymal Plasticity using Organelle Dynamics.

Corresponding Author: Professor Jonas Fuxe

Version 0:

Reviewer comments:

Reviewer #1

(Remarks to the Author)

The paper, "A Morphology-Based Machine Learning Model for Scoring Epithelial-Mesenchymal Plasticity using Organelle Dynamics," by Slager et al., introduces a novel morphology-based machine learning method to predict EMT status by organelle morphology. The idea is interesting, and the paper is well-organized. However, there are several limitations. The deep learning method used in this work is just conventional without significant innovation. Biologically speaking, the data collection and validation are not rigorous. There are also some relevant prior studies which are not compared and discussed. It is recommended that the paper should be reconsidered after significant revision. Detailed comments are listed below:

1. As suggested by the authors in the introduction, EMT can have complicated molecular features depending on cell lines and contexts. It is recommended that the authors can perform some molecular analysis on the samples used in this study to verify whether and which EMT markers are truly altered with the stimulation.
2. TGFB is just one way to induce EMT. It is recommended that the authors can test at least one additional strategy to induce EMT for comparison.
3. It is recommended that the authors can quantify the accuracy of single-cell segmentation, especially when cells are crowded. Some mistakes in cell segmentation can significantly affect the morphological features.
4. It is recommended that the authors can discuss cellular heterogeneity based on the data. Are all cells induced to be EMT-like?
5. The cell status transition in the Fig.3b might be just explained by the increase of cell crowdedness over time. When cells are getting more crowded, the morphology can naturally change. It might not be caused by EMT or MET. It is recommended that the authors should carefully consider crowdedness effects, especially when culturing highly proliferative MDA-MB-231 cells for 8 days.
6. There are some prior articles related to cellular morphology under EMT with TGFB treatment (e.g., PMID: 32917609). It is recommended that the authors can discuss and compare the discovery with the prior work.
7. Cell motility is highly correlated with EMT. It is recommended that the authors can also discuss whether the observed EMT morphology is correlated with motile cell morphology in prior articles (PMID: 18497816, 30420987, 39739899).

Reviewer #2

(Remarks to the Author)

The authors have developed a cell morphology based ML model to predict the state of EMT/MET in cells leveraging the information about organelle shape changes during EMT/MET. By collecting high-throughput microscopy data and Cell Painting assay for multiple cell lines treated with TGFB1, they develop a morphology-based EMT score. Overall, the study is

well-designed but needs to address few points:

1. Given that identifying the unique morphological features of hybrid E/M states is a key goal here, what specific criteria were used to classify individual cells into hybrid? Also, please clarify exact morphological variables being used as output and for training the ML model.
2. At least in the biochemical state space, EMT can proceed through multiple trajectories as shown in previous reports given below. Do the authors note different trajectories in morphological state space as well for the same cell line? Line 378 mentions PAGA, but details are missing. What is the reference for the method, and how many trajectories did it reveal?
<https://pubmed.ncbi.nlm.nih.gov/39093947/>
<https://pubmed.ncbi.nlm.nih.gov/35188459/>
3. Fig 6A shows that EMT score remained unchanged for the 2-day reversal timepoint, but not for 4-day and 8-day reversal timepoints. Similar observations about the delayed recovery from EMT at a biochemical level have been reported that the authors should discuss (<https://royalsocietypublishing.org/doi/full/10.1098/rsif.2022.0627>). Moreover, what is the explanation for 8-day reversal EMT score being less than the control/untreated cells?
4. Line 406: How are the weights being distributed for the different states? Is there a justification for choosing those specific numbers?
5. The authors should discuss their results in the context of other previous reports investigating dynamic changes in shape during EMT:
<https://pubmed.ncbi.nlm.nih.gov/31247884/>
<https://www.nature.com/articles/s41598-023-48279-y>
6. A recent study showed via scRNA-seq analysis that while luminal cell lines are predominantly epithelial, basal cell lines can be more heterogeneous in terms of cells spread across the phenotypic spectrum ([https://www.cell.com/science/fulltext/S2589-0042\(24\)01341-5](https://www.cell.com/science/fulltext/S2589-0042(24)01341-5)). Could the authors also include the variance (in addition to mean) of scores in these cell lines for the data shown in Fig 6B?

Reviewer #3

(Remarks to the Author)

Epithelial-mesenchymal transition (EMT) is a developmental process involving multiple pathways, resulting in a panel of functional responses including phenotype plasticity and cytoskeleton reorganization. The transition to a mesenchymal phenotype may be partial and only include some of these responses. In this manuscript, authors describe a new method to quantify the extent of EMT expressed by a cell responding to an induction based on image analysis. They use a panel of classic mammary epithelial cells lines used frequently in EMT studies and an induction by TGF β , one of the best described EMT-induction method. They use a "Cell painting assay" to label organelles and analyze morphology data using high-throughput microscopy and machine-learning to obtain an EMT score from control and treated cells. They offer this scoring method as a pre-clinical tool to evaluate treatment and get a deeper understanding on cell plasticity and drug response during cancer progression.

The manuscript is clear and well written. The rationale is solid: there is an interest in quantifying EMT status in cancer cells and drug-linked modulation. The study is based on the organelle dynamics occurring during EMT, assuming that the partial EMT observed during cancer progression will be reflected by limited organelle dynamics that should keep proportional somehow to the intensity of the EMT induction. This notion of a linear process is arguable and EMT-like plasticity may take different aspects in distinct cell types. Keeping this limit in mind, the results appear sturdy and support at least qualitatively the conclusions. Some points remain unclear, like the interest of the method compared to a functional characterization of cell phenotype based on cytoskeletal, cell adhesion structures, cell polarity and motility using classic immunolocalization in 2D or 3D models. This should be convincingly answered to give more impact to the manuscript.

Major points

TGF β is a "classic" EMT inducer. However, it is not the only pathway to lead to EMT. It would have been relevant to test at least another growth factor or distinct pathway (stiffness, hypoxia...) to check how general the method can be. One unique and high dose (10ng/ml) of TGF β is used here. This dose is expected to induce a strong and full EMT with time, as suggested by the images from Figs 2 and S1. The EMT score calculation is based on a time-course series of data, providing a good range of response (about 1 to 6) using this full range of EMT stages through time. But cell lines, reflecting cancer cell partial phenotypes, do not provide such a dynamic range: the archetype cancer epithelial cell line MCF7 has an EMT score barely 1.5 time lower than the archetype cancer mesenchymal cell line MDA-MB-231. This limits the value of the method as a preclinical measurement tool of EMT.

Minor points

Overall, the method is not simple and requires specific cell culture, staining and image analysis hardware supporting spinning disk fluorescence microscopy with a multiband filter wheel. It requires the cells to grow on a 2D substrate. It does not appear to be appropriate for primary cancer cells.

Line 106: cells are described as elongated which is not clear in figure 2, but visible in supplementary figure S1.

Line 112: The list of the 2,600 morphological features should be accessible in supplementary data

Line 113: It is not clear how the 90 profiles are designed.

Lines 112-118: The selection starts with 2,600 features (line 112), then removing 2,060 features (line 118) should leave 540 features. But in Figure 3 there are 2,638 features reduced to 405 features, what are the correct numbers?
Line 113: Figure 3 mentions 1620 aggregated, but only 90 are mentioned in the text line 113. The process is not clear.
Line 169: the calculation method is not clear : it could show an example of the calcul, P_n calculated in figure 5b ? and W_n number of wells?
Line 206: Numerous basal A and B cell lines are depicted, but there is no result analysis.
Line 405: How and where was calculated exactly the progression score?
Lign 406: Is P(Control) estimated on 405 features or from the 15 last features selected? and how were defined the predefined score?
Lign 424: Why choosing to do the statistical analysis in GraphPad Prism while the EMT score was done on Python and R?
Lign 427: Access to these data is not permitted without revealing the reviewer identity/affiliation
Figure 5: Legend is confusing. Which image was created with BioRender.com and why?

Version 1:

Reviewer comments:

Reviewer #1

(Remarks to the Author)

The authors addressed the comments from the reviewers.

Reviewer #2

(Remarks to the Author)

The authors have satisfactorily addressed my comments.

Reviewer #3

(Remarks to the Author)

Authors have answered satisfactorily to our questions.

Rebuttal letter on MS COMMSBIO-25-1988-T

Reviewers' comments:

Reviewer #1 (Remarks to the Author):

The paper, "A Morphology-Based Machine Learning Model for Scoring Epithelial-Mesenchymal Plasticity using Organelle Dynamics," by Slager et al., introduces a novel morphology-based machine learning method to predict EMT status by organelle morphology. The idea is interesting, and the paper is well-organized. However, there are several limitations. The deep learning method used in this work is just conventional without significant innovation. Biologically speaking, the data collection and validation are not rigorous. There are also some relevant prior studies which are not compared and discussed. It is recommended that the paper should be reconsidered after significant revision. Detailed comments are listed below:

Response: We thank the reviewer for valuable and insightful comments on our manuscript. Below, please find a point-by-point response to the comments.

1. As suggested by the authors in the introduction, EMT can have complicated molecular features depending on cell lines and contexts. It is recommended that the authors can perform some molecular analysis on the samples used in this study to verify whether and which EMT markers are truly altered with the stimulation.

Response: Yes, we agree and have performed imaging and western blot analysis on our samples to verify the induction of EMT in our NMuMG model (New Supplementary Figure S1a-c). Text describing these data have been included in the Results (lines 100-104).

2. TGFB is just one way to induce EMT. It is recommended that the authors can test at least one additional strategy to induce EMT for comparison.

Response: Yes, we agree and performed a series of new experiments to test if the model could be used to score EMT in hypoxia-induced EMT in human A549 lung cancer cells. The results from these experiments confirmed this and are presented in new figures (Fig. 7 and Supplementary Figure S8), described in the Results (lines 219-227).

3. It is recommended that the authors can quantify the accuracy of single-cell segmentation, especially when cells are crowded. Some mistakes in cell segmentation can significantly affect the morphological features.

Response: Yes, we agree and appreciate that the reviewer highlighted this point. In the revised version of the manuscript, we have clarified how measurements were taken to assure that the cells were accurately segmented (Method section; lines 405-407)

4. It is recommended that the authors can discuss cellular heterogeneity based on the data. Are all cells induced to be EMT-like?

Response: Yes, we think this a good suggestion and have included a new section of the Discussion in which our data is discussed related to cellular heterogeneity (lines: 268-285)

5. The cell status transition in the Fig.3b might be just explained by the increase of cell crowdedness over time. When cells are getting more crowded, the morphology can naturally change. It might not be caused by EMT or MET. It is recommended that the authors should carefully consider crowdedness effects, especially when culturing highly proliferative MDA-MB-231 cells for 8 days.

Response: We appreciate that the reviewer raised this important point. It highlights the fact that although it was an integral part of the design of the experiments to make sure that we were not analyzing cells at different cell confluency, we did not describe this thoroughly enough in the original submission. In the revised manuscript, we have included a more detailed description of how cells were regularly split and reseeded over the time course of the experiments, and that at 24 h before the cell painting was performed, they were detached and seeded with the same number of cells per well (Method section lines: 352-362)

6. There are some prior articles related to cellular morphology under EMT with TGFB treatment (e.g., PMID: 32917609). It is recommended that the authors can discuss and compare the discovery with the prior work.

Response: Yes, we agree and have included a new section of the discussion where this is discussed (lines: 249-252)

7. Cell motility is highly correlated with EMT. It is recommended that the authors can also discuss whether the observed EMT morphology is correlated with motile cell morphology in prior articles (PMID: 18497816, 30420987, 39739899).

Response: Yes, we agree and have included this in a new section of the discussion (lines: 259-261)

Reviewer #2 (Remarks to the Author):

The authors have developed a cell morphology based ML model to predict the state of EMT/MET in cells leveraging the information about organelle shape changes during EMT/MET. By collecting high-throughput microscopy data and Cell Painting assay for multiple cell lines treated with TGF β 1, they develop a morphology-based EMT score. Overall, the study is well-designed but needs to address few points:

1. Given that identifying the unique morphological features of hybrid E/M states is a key goal here, what specific criteria were used to classify individual cells into hybrid? Also, please clarify exact morphological variables being used as output and for training the ML model.

Response: We agree that it was needed to describe this better. We have included a section in the Results (lines: 110-119; 125-126; 150-157) where we clarify how many and which morphological variables were obtained as an output and used to train the model. We also clarify that we used aggregated profiles rather than single cells for the EMT scoring of each time point (Results, line 116). Furthermore, we have added a section in the Discussion (lines: 269-273), where we clarify that the hybrid state was classified based on aggregated profiles rather than single cell-based scoring.

2. At least in the biochemical state space, EMT can proceed through multiple trajectories as shown in previous reports given below. Do the authors note different trajectories in morphological state space as well for the same cell line? Line 378 mentions PAGA, but details are missing. What is the reference for the method, and how many trajectories did it reveal?

<https://pubmed.ncbi.nlm.nih.gov/39093947/>

<https://pubmed.ncbi.nlm.nih.gov/35188459/>

Response: Thank you for pointing out the literature showing that EMT can proceed along multiple biochemical trajectories and asking whether we observe analogous trajectories in morphological state space. In the revised manuscript, we have included a new section in the Discussion (lines: 268-285), where we discuss this in relation to the mentioned papers. Furthermore, we have included a reference to PAGA, and clarified the reason for using this method (Method section, lines: 428-433).

3. Fig 6A shows that EMT score remained unchanged for the 2-day reversal timepoint, but not for 4-day and 8-day reversal timepoints. Similar observations about the delayed recovery from EMT at a biochemical level have been reported that the authors should discuss

(<https://royalsocietypublishing.org/doi/full/10.1098/rsif.2022.0627>). Moreover, what is the explanation for 8-day reversal EMT score being less than the control/untreated cells?

Response: We have included a paragraph discussing this (lines: 280-285)

4. Line 406: How are the weights being distributed for the different states? Is there a justification for choosing those specific numbers?

Response: We have clarified this in the Methods (lines: 459-464) and in Fig legend of Fig. 5c.

5. The authors should discuss their results in the context of other previous reports investigating dynamic changes in shape during EMT:

<https://pubmed.ncbi.nlm.nih.gov/31247884/>

<https://www.nature.com/articles/s41598-023-48279-y>

Response: We have included a paragraph in the Discussion (lines: 262-269)

6. A recent study showed via scRNA-seq analysis that while luminal cell lines are predominantly epithelial, basal cell lines can be more heterogeneous in terms of cells spread across the phenotypic spectrum ([https://www.cell.com/iscience/fulltext/S2589-0042\(24\)01341-5](https://www.cell.com/iscience/fulltext/S2589-0042(24)01341-5)). Could the authors also include the variance (in addition to mean) of scores in these cell lines for the data shown in Fig 6B?

Response: Thank you for this comment. We have added the variance in Fig. 6C and clarified the box plot indicators (IQR) in the figure legend of Fig. 6B.

Reviewer #3 (Remarks to the Author):

Epithelial-mesenchymal transition (EMT) is a developmental process involving multiple pathways, resulting in a panel of functional responses including phenotype plasticity and cytoskeleton reorganization. The transition to a mesenchymal phenotype may be partial and only include some of these responses. In this manuscript, authors describe a new method to quantify the extent of EMT expressed by a cell responding to an induction based on image analysis. They use a panel of classic mammary epithelial cells lines used frequently in EMT studies and an induction by TGF β , one of the best described EMT-induction method. They use a "Cell painting assay" to label organelles and analyze morphology data using high-throughput microscopy and machine-learning to obtain an EMT score from control and treated cells. They offer this scoring method as a pre-clinical tool to evaluate treatment and get a deeper understanding on cell plasticity and drug response during cancer progression. The manuscript is clear and well written. The rationale is solid: there is an interest in quantifying EMT status in cancer cells and drug-linked modulation. The study is based on the organelle dynamics occurring during EMT, assuming that the partial EMT observed during cancer progression will be reflected by limited organelle dynamics that should keep proportional somehow to the intensity of the EMT induction. This notion of a linear process is arguable and EMT-like plasticity may take different aspects in distinct cell types. Keeping this limit in mind, the results appear sturdy and support at least qualitatively the conclusions. Some points remain unclear, like the interest of the method compared to a functional characterization of cell phenotype based on cytoskeletal, cell adhesion structures, cell polarity and motility using classic immunolocalization in 2D or 3D models. This should be convincingly answered to give more impact to the manuscript.

Major points

TGF β is a "classic" EMT inducer. However, it is not the only pathway to lead to EMT. It would have been relevant to test at least another growth factor or distinct pathway (stiffness, hypoxia...) to check how general the method can be.

One unique and high dose (10ng/ml) of TGF β is used here. This dose is expected to induce a strong and full EMT with time, as suggested by the images from Figs 2 and S1. The EMT score calculation is based on a time-course series of data, providing a good range of response (about 1 to 6) using this full range of EMT stages through time. But cell lines, reflecting cancer cell partial phenotypes, do not provide such a dynamic range: the archetype cancer epithelial cell line MCF7 has an EMT score barely 1.5 time lower than the archetype cancer mesenchymal cell line MDA-MB-231. This limits the value of the method as a preclinical measurement tool of EMT.

Response: We agree with the reviewer that it is important to validate the robustness and generalizability of the method in another context. We have therefore added a whole new set of data presented in the new Fig. 7, showing that the model is able to score hypoxia-induced EMT in human A549 lung cancer cells.

Moreover, we agree that many cancer cell lines have partial EMT phenotypes and not all will be possible to separate from each other using the scoring method. To explore this, we generated a new figure (Fig. 8), where we plotted EMT scores from all cell lines and conditions in the same graph. Although the difference between MCF7 cells and MDA-MB-231 cells may seem low (50%), we noted that it is significantly higher than the difference in the expression of the EMT hallmark genes (200 genes), which was only 2%. Moreover, MCF7 cells, although often referred to as an epithelial/luminal breast cancer cell line, are known for their heterogeneity and for constituting various subpopulations (reference included in the Discussion), which was also reflected in our data (Fig. 6B). In

addition to the new 8, we have included a new part of the Discussion where we discuss this (lines: 315-323).

Minor points

Overall, the method is not simple and requires specific cell culture, staining and image analysis hardware supporting spinning disk fluorescence microscopy with a multiband filter wheel. It requires the cells to grow on a 2D substrate. It does not appear to be appropriate for primary cancer cells.

Response: We agree and talk about this in the revised Discussion (lines: 303-307).

Line 106: cells are described as elongated which is not clear in figure 2, but visible in supplementary figure S1.

Response: The images in Fig. 2 show staining with the different organelle markers and therefore display different parts of the cells. We believe that cellular elongation is clearly visible in the phalloidin and wheat germ agglutinin channels, which stain the cytoskeleton and plasma membrane, respectively. To further demonstrate cellular elongation we have added a new Fig.S1a with brightfield images of NMuMG cells at each of the time points after TGF- β 1 stimulation.

Line 112: The list of the 2,600 morphological features should be accessible in supplementary data

Response: We thank the reviewer for the suggestion. We have now included a new supplementary table (Supplementary Table 1) containing the list of all the 2608 features.

Line 113: It is not clear how the 90 profiles are designed.

Response: We have clarified this in the figure legend of Fig. 3a.

Lines 112-118: The selection starts with 2,600 features (line 112), then removing 2,060 features (line 118) should leave 540 features. But in Figure 3 there are 2,638 features reduced to 405 features, what are the correct numbers?

Response: We thank the reviewer for pointing out our typos – we have now replaced the numbers with the correct ones both in the Results (lines: 125-126) and in Fig. 3 (figure + legend).

Line 113: Figure 3 mentions 1620 aggregated, but only 90 are mentioned in the text line 113. The process is not clear.

Response: We thank the reviewer for the comment. We agree that the process is unclear, and we have better described the numbers in the figure legend (Fig 3a).

Line 169: the calculation method is not clear : it could show an example of the calcul, P_n calculated in figure 5b ? and W_n number of wells?

Response: We thank the reviewer for the comment. We have clarified the calculation process both in the figure legend (Fig. 5B) and in the methods section (lines 459-464).

Line 206: Numerous basal A and B cell lines are depicted, but there is no result analysis.

Response: Thank you for pointing this out. We have revised the Results section (lines: 205-216) to include the result analysis.

Line 405: How and where was calculated exactly the progression score?

Response: We thank the reviewer for the comment. We have now included this in the method (lines 459-464).

Lign 406: Is P(Control) estimated on 405 features or from the 15 last features selected? and how were defined the predefined score?

Response: We thank the reviewer to highlight this and have clarified it in the Method section (lines: 451-464).

Lign 424: Why choosing to do the statistical analysis in GraphPad Prism while the EMT score was done on Python and R?

Response: We thank the reviewer for pointing this out and understand that it could lead to misunderstanding. We have now clarified in the Methods (line 495) that statistical analyses were performed in parallel using both Python and GraphPad Prism to ensure consistency. GraphPad Prism was used only for plotting, as it provides greater flexibility for graphical customization and improved presentation quality.

Lign 427: Access to these data is not permitted without revealing the reviewer identity/affiliation

Response: We have attached the links below.

CP_INT_03_Normoxia_nd2

<https://figshare.com/s/3a504418809ccc61bc56>

CP_INT_03_Hypoxia_nd2

<https://figshare.com/s/9dd4d7facc5d34ea4250>

CP020-2_nd2

<https://figshare.com/s/1422fed1469bb03aff92>

CP019_nd2

<https://figshare.com/s/ef586bf3a4ce54913c90>

CP018_nd2

<https://figshare.com/s/ca8d010454c6c48743e8>

CP017_nd2

<https://figshare.com/s/1980629a825f85a889df>

CP016_nd2

<https://figshare.com/s/b70caa200484ef54af34>

CP015_nd2

<https://figshare.com/s/3028849fe3c85eedd1ef>

Figure 5: Legend is confusing. Which image was created with BioRender.com and why?

Response: We have clarified in the figure legend (5C) which panel was created with biorender.